# Presence of *Mycoplasma bovis* in Bulk Tank Milk and Associated Risk Factor Analysis in Serbian Dairy Farms

**DOI:** 10.3390/pathogens13040302

**Published:** 2024-04-06

**Authors:** Milan Ninković, Vesna Milićević, Sonja Radojičić, Dejan Bugarski, Nataša Stević

**Affiliations:** 1Scientific Institute of Veterinary Medicine of Serbia, Janisa Janulisa 14, 11000 Belgrade, Serbia; vesna.milicevic@nivs.rs; 2Faculty of Veterinary Medicine, University of Belgrade, Bulevar Oslobodenja 18, 11000 Belgrade, Serbia; sonjar@vet.bg.ac.rs (S.R.); natasas@vet.bg.ac.rs (N.S.); 3Scientific Veterinary Institute Novi Sad, Rumenački put 20, 21113 Novi Sad, Serbia; dejan@niv.ns.ac.rs

**Keywords:** *Mycoplasma bovis*, bulk tank milk, risk factors, Serbia

## Abstract

*Mycoplasma bovis* (*M. bovis*) is a significant pathogen responsible for highly transmissible mastitis in cattle globally. It primarily spreads through colostrum, milk, and semen. Cows with persistent infections act as carriers, intermittently releasing the pathogen, making their milk a pivotal factor in infection transmission. Given the limited seroprevalence surveys in Serbia, this study aimed to detect *M. bovis* presence in bulk tank milk (BTM), determine route shedding, and evaluate infection risks. BTM samples were collected from 115 dairy farms across Serbia, with *M. bovis* DNA detected in 11 out of the 115 samples by real-time PCR. Additionally, *M. bovis* was detected in 1.30% of nasal swabs sampled from apparently healthy animals. A univariate analysis of the risk factors associated with *M. bovis* presence in the BTM samples revealed correlations with factors such as the breed, farm seropositivity, pre-milking and post-milking disinfection practices, farm type, cow population, milk yield, number of cows in the BTM samples, and parity. Seropositive farms exhibited the highest likelihood of *M. bovis* presence in milk. Moreover, pre- and post-milking disinfection practices and highly productive cows yielding over 8000 L of milk were identified as risk factors for PCR-positive BTM. In a multivariable mixed regression analysis, a risk factor for the presence of *M. bovis* infection in the BTM sample was the Holstein breed. These findings underscore a relatively high prevalence of *M. bovis* in BTM within Serbian dairy farms, suggesting a potential risk for *M. bovis* spreading through milk and oral route of calves’ infection.

## 1. Introduction

*Mycoplasma bovis* (*M. bovis*) is one of the most notable pathogens that cause highly contagious mastitis in cattle worldwide [1]. Contagious mastitis is deemed a significant contributor to economic downturns in the dairy industry because it leads to decreased milk production and compromised milk quality [2]. In addition to mastitis, *M. bovis* causes bronchopneumonia, arthritis, otitis, and keratoconjunctivitis [3]. 

Mycoplasmas are the smallest prokaryotic bacteria; as microorganisms with a small genome, mycoplasmas have limited physiological metabolic activities dependent on the host and occupy an extra- or intracellular position in the organism [4]. Mycoplasmas have specialised survival mechanisms, such as immune mimicry, by which mycoplasmas avoid the body’s immune response, preventing their elimination and enabling the development of a chronic form of the disease [5]. In addition, *M. bovis* possesses numerous factors that lead to immunosuppression, whether they block the production of lymphocytes and phagocytes [6].

The main sources of *M. bovis* are colostrum, milk, and semen, whereas infection can also be disseminated through airborne transmission or intrauterine routes [7]. However, several factors contribute to disease development [8], such as impaired immunity caused by starvation, transport, low temperatures, and other diseases [9]. Mastitis induced by *M. bovis* can appear either clinically or sub-clinically [10], with the acute course progressing into a chronic one. Cows with chronic infections serve as carriers, intermittently shedding the pathogen. Therefore, their milk is considered crucial for the spread of infection [11]. Current approaches for *M. bovis* control such as BTM testing can be used for *M. bovis* screening [12,13]. Considering the intermittent shedding of *M. bovis* by carriers, their identification can be demanding [14]. Therefore, repeated sampling to increase the likelihood of *M. bovis* detection in both clinically and sub-clinically infected animals is required [15,16]. Furthermore, cows displaying elevated milk somatic cell counts, indicative of subclinical mastitis, should be identified and tested. Hence, the primary risk factor for *M. bovis* introduction into a herd is the introduction of asymptomatic carriers [17], underscoring the importance of implementing biosecurity measures such as isolating all cattle before their entry into a new herd. 

Infections caused by *M. bovis* bacteria can be diagnosed using classical bacteriological methods, serological tests, and molecular methods. Combinations of different samples and analysis methods are key in diagnostics, especially in detecting asymptomatic carriers in herds. Although culturing is a gold standard, real-time PCR has an advantage given its diagnostic performances [18]. However, for the most accurate PCR results, sample enrichment is usually conducted prior to the process [19]. With no efficient vaccine and medical treatment, *M. bovis* infection can be controlled using different approaches, such as ‘test and slaughter’ or ‘test and segregate’ [7]. In Serbia, few seroprevalence surveys have been conducted across different categories of cattle [20,21,22]. The aim of this study was to determine the presence of *M. bovis* in BTM samples in Serbia and route of shedding and assess the risk for *M. bovis* infection.

## 2. Materials and Methods

### 2.1. Study Design and Sampling

The survey was conducted on 115 dairy farms randomly selected throughout Serbia in 2022. The total number of dairy cow farms to be sampled (115) was determined with a 10% expected prevalence according to the results of Vojinović et al. [22], 5% absolute precision, and 95% confidence interval (CI) in reference to the total number of these farms in Serbia based on the data from the Statistical Office of the Republic of Serbia [23]. The farms were categorised as small-, medium-, or large-scale farms according to the number of dairy cows (<20, 20–200, or >201, respectively). General data such as the breed of cows, herd structure, number of cows, and milk yield were recorded during the farm visits. Data were collected using a structured risk factor questionnaire with responses given to farm owners or veterinarians. We collected information on herd structure and husbandry management via a structured questionnaire. All the questionnaires were sent via email. Milk production was calculated as the mean production per cow for the past 12 months. In addition, to define the risk factors, we performed blood sampling and took nasal swabs from healthy cows in the examined farms. A two-stage random sampling process from healthy cows was carried out in this study. Dairy cow farms were selected in the first stage, and cows within flocks were selected in the second stage. In total, blood samples and nasal swabs were collected from 307 cows from 61 herds. The representative number of nasal swabs and blood samples—up to 10 samples per farm—was taken to evaluate the excretion of *M. bovis* and serostatus of each farm. Blood samples (10 mL) were collected from the selected cattle from the coccygeal vein and stored using BD-Vacutainer^®^ 10 mL tubes. Sera were centrifuged from the clotted blood in the collection tubes at 1000 g for 5 min and stored at −20 °C until the analysis. The milk samples were collected from bulk tanks in 50 mL tubes, and all the samples were stored in cold storage and immediately sent to the laboratory for diagnostic analyses. 

### 2.2. Risk Factors

A structured risk factor questionnaire with 16 variables was given to the farm owners to fill in and send back via email. The questionnaire is shown in Appendix A. A total of 61 answered questionnaires were returned—which represents 53% participation—and included in a further risk analysis. Potential risk factors were tested univariably for their association with the outcome variable. Of the 16 recorded variables, 4 were continuous (the number of cows at the farm, milk yield, parity, number of cows in terms of BTM). The presence of *M. bovis* mastitis as a variable was excluded from the analysis, while the presence of mycoplasma infection was not previously confirmed on the examined farms. The remaining 11 categorical variables were as follows: breed (HF/SIM), positive nasal swab (yes/no), bacterial mastitis (yes/no), seropositive farm (yes/no), disinfection before milking (yes/no), disinfection after milking (yes/no), type of farm (family farm/corporate farm), type of milking (machine/manual), type of holding (tie stall/free stall), overcrowding of stall (yes/no), and inadequate ambition condition (yes/no). The farm was categorised as a family farm if the total number of animals in the herd was up to 20 cows and a commercial farm if it was above 20 cows. 

### 2.3. Antibody Detection

An indirect enzyme-linked immunosorbent assay (ELISA) kit for the detection of antibodies against *M. bovis* (IDvet Screen *Mycoplasma bovis* Indirect, IDVet, Grabels, France) was used according to the manufacturer’s instructions. The farm was considered *M. bovis*-positive if one or more positive cows were detected.).

### 2.4. Molecular Detection of M. bovis DNA

Bulk tank samples were centrifuged at 1,000 g for 10 min. The supernatants were discarded, and the resuspended pellets were used for DNA extraction via the commercial IndiSpin Pathogen Kit (Indical, Leipzig, Germany), following the manufacturer’s instructions. 

The nasal swabs, immersed in 0.9 mL of PBS and vigorously vortexed, were also subjected to DNA extraction. To detect the *M. bovis* genome, a real-time PCR protocol using primers previously described [24] for the amplification of the oppD gene and Luna^®^ Universal qPCR Master Mix (NEB, Ipswich, MA, USA) were used. The samples with Ct values of ≤40 were considered positive. 

### 2.5. Statistical Analysis

The obtained results were analysed using descriptive statistics methods. A univariable analysis was carried out, and the associations between the outcome and the dichotomous variables were evaluated using Fisher’s exact test. The association between the outcome and the continuous variables was analysed using Student’s *t*-test. Risk factors of *M. bovis* infection were studied through a mixed effects logistic regression model. CIs were estimated with the exact method [25]. Significant p-values were set to *p* < 0.05. The statistical analyses were performed using JASP (JASP Team, version 0.16.0).

## 3. Results

Herd distribution based on cow breed was as follows: Simmental = 74.8% and Holstein = 25.2%. The mean herd size of the sampled farms was 83.122 (SD = 182.88) and ranged from 1 to 1000 (Table 1). An overview of the results of the analysed samples in this study is shown in Table 1. Out of 115 dairy cow farms, *M. bovis* DNA was detected in 9.57% (95% CI: 4.87–16.47%) (11/115) of the BTM samples. *M. bovis* DNA was detected in 1.30% (95% CI: 0.36–3.30%) (4/307) of the nasal swabs collected, as determined by real-time PCR. PCR-positive cows based on nasal swabs were detected in 4.92% (95% CI: 1.03–13.71%) (3/61) of the dairy farms. Among the 61 sampled dairy farms, ELISA positivity was detected in 22.95% (95% CI: 13.15–35.50%) (14/61) of the farms. The overall seroprevalence of *M. bovis* in 307 dairy cows was 37.79% (95% CI: 31.35–43.07%). Out of the 115 distributed questionnaires concerning the risk factors, 61 were returned (53%) and utilised for risk assessment.

Concerning the distribution of herd size, the results are presented in Table 2. *M. bovis* DNA was most often detected in herds with more than 201 cows in farms (71.43%). 

Concerning the distribution of milk yield, the results are presented in Table 3. *M. bovis* DNA was most commonly detected in farms with milk yield above 8000 L (38.89%). 

The Student’s *t*-test results for potential risk factors based on continuous variables on dairy cow farms are presented in Table 4. The average number of cows in terms of BTM samples for positive farms was 216.09, ranging from 34 to 600, while the average number of cows in terms of BTM samples for negative farms was 54.95. Thus, the evident risk factors for *M. bovis* infection were the number of cows at farms, milk yield, the number of cows in terms of BTM samples, and parity, as shown in Table 4.

The Fisher test results for the potential risk of categorical variables on dairy cow farms are shown in Table 5. The recognised risk factors for *M. bovis* infection were the breed, seropositive farm status, disinfection before milking, disinfection after milking, and type of farm. 

External risk factors with *p* < 0.2 in the univariable analysis and confounding factors were included in a final regression model by the enter method. Biologically meaningful interactions were tested for as well. Multicollinearity diagnostics was performed by the inverse of the Variance Inflation Factor where large values indicate multicollinearity. Two variables, all of which were not statistically correlated, remained in the final model. After a collinearity check, the factors of interest remained overcrowding and breed. The goodness of analysis is measured by McFadden’s R2 value of 0.47, with 91.803% accuracy of correctly predicted outcomes in the confusion matrix. In the breed sanction, HF breeds’ odds are 65.069 times more than Simmental for *M. bovis*-positive BTM samples (Table 6).

The findings revealed that dairy cows producing more than 8000 L of milk were at a 9.43-fold greater risk of developing *M. bovis* mastitis compared to cows with milk yields below 8000 l per lactation. This difference was statistically significant (*p* < 0.0001), as shown in Table 7.

Our findings indicate that an increasing herd size significantly elevates the risk of detecting *M. bovis* DNA in BTM samples (*p* < 0.001). Specifically, herds with 51 or more cows have a 16.02-fold higher risk of *M. bovis* DNA presence compared to herds with less than 50 cows. Similarly, herds with 101 or more cows have an 11.94-fold higher risk, herds with 201 or more cows have a 13.70-fold higher risk, and herds with 301 or more cows have a 10.30-fold higher risk of *M. bovis* DNA detection compared to their respective smaller counterparts (Table 8).

## 4. Discussion

This study has revealed novel perspectives on the occurrence of *M. bovis* in dairy cow herds in Serbia. *M. bovis* DNA was identified in 11 BTM samples (9.67%), consistent with findings from other countries [12,26,27]. In Denmark, however, only 2% of the BTM samples tested positive for *M. bovis* DNA [28]. These results indicated that the presence of infection in Serbian farms may be another important reason for the relatively high prevalence of *M. bovis* in the examined farms, which may be caused by intramammary infection. A positive BTM PCR result indicates the presence of *M. bovis* on farms, and then based on the somatic cell count, the cows should be able to be selectively tested for the presence of *M. bovis* in milk. This could be explained by mastitis caused by *M. bovis*, which leads to an increase in the number of somatic cells and dropping milk yield [29]. Daily milk losses associated with the occurrence of *M. bovis* subclinical intramammary infection were an average of 3.0 kg lower with the decreased content of milk components [2].

Our PCR results from nasal swabs indicated the presence of *M. bovis* infection and excretion via nasal secretions. This finding is also supported by similar results in a study by Moore et al. [30]. Based on our results, we did not find a farm with the simultaneous excretion of *M. bovis* through milk and nasal secretions. Contrary to our results, Garcia-Galan et al. [31] reported presence of *M. bovis* in 32% of healthy animals considered asymptomatic carriers. The reduced detection of *M. bovis* in nasal swabs in our study might be due to intermittent shedding or shedding below the detection threshold, given the absence of respiratory disease or symptoms in the animals. Consequently, these findings suggest the presence of asymptomatic carriers, posing a risk for the emergence of respiratory diseases within herds [15]. In this study, we also found the seroprevalence of *M. bovis* to be 37.79%, in accordance with the results of Gogoi-Tiwari et al. [32]. That being said, among the main limitations of this study are the limited number of farms, the wide variations in terms of herd size, the management practices, and representing the majority of farms in the Republic of Serbia. 

PCR proves to be an effective and valuable tool for detecting *M. bovis* in BTM samples, enabling screening for infected dairy cows shedding the pathogen. Consequently, it has been recommended for integration into surveillance programs. The application of the PCR method to detect the presence of *M. bovis* on cow farms is significant for the early diagnosis and prevention of a further spread of *M. bovis* within the herd compared to the classic bacteriological method. A positive BTM PCR can be caused by only a few shedding animals and may indicate the intermittent or persistent excretion of *M. bovis* as well as excretion via other routes. The LOD PCR for the detection of *M. bovis* is between 10 and 240 cfu/mL in milk [16]. However, factors such as the presence of and intermittent shedding of *M. bovis* also complicate the diagnosis, which is why repeated sampling is recommended to increase the detection of *M. bovis* [15,16,33]. 

After a univariable analysis of 16 variables, we identified risk factors associated with BTM positive results for *M. bovis*, including the number of cows at the farm, milk yield, number of cows in terms of BTM samples, parity, breed, seropositivity, disinfection before milking, disinfection after milking, and type of farm. 

The most important risk factors for the spread of *M. bovis* among herds are the herd size, semen, and purchase of heads of an unknown status [34,35]. The arrival of infected heads among cows is a source for the spread of *M. bovis* during their stay at purchase points as well as during transport and arrival at the herd.

Based on our obtained results, herd size was the main risk factor for finding *M. bovis* in BTM samples, in accordance with previous studies [34,36]. Comparable results were noted in Japan, where larger herd sizes and corporate farms exhibited a heightened likelihood of testing positive for *M. bovis* [36]. Cows on large farms have more contact with other animals; this is accompanied by stress, and moving during the production process to different production stages may lead to significant contact with other cows. Previous studies have also identified housing as a risk factor [36], contrary to our results as we did not identify this to be a statistically significant risk factor for the presence of *M. bovis* in BTM samples. 

Furthermore, herds testing positive for antibodies against *M. bovis* were 55 times more likely to have a positive result for *M. bovis* in BTM samples (*p* < 0.001). Dairy cows are frequently housed in large systems with intensive production, facilitating the spread of *M. bovis* among animals. Within larger dairy herds, the increased number of interactions may heighten the risk of exposure to the pathogen from an infected animal [37]. This study revealed that the Holstein Friesian breed statistically exhibited greater susceptibility to *M. bovis* infection, as in previous research conducted in western Australia [32]. Regarding risk factors, the multivariable analysis revealed that the Holstein breed had about 65 times greater odds than the Simmental breed for PCR-positive BTM samples. The findings agree with the report by Pires et al. [38], who reported that the Holstein breed has more than 70 times greater odds compared to crossbreed cows. 

Parity was also recognised as a risk factor for PCR-positive BTM samples. There was a significant difference in detecting *M. bovis* in BTM samples from younger cows (≤4 lactations) compared with older (≥4 lactations) dairy cows. A possible cause of this effect can include having a lower number of lactations, which is conditioned by metabolic, reproductive, and hormonal conditions. By redirecting metabolism and potentiating the milk production process, physiological homeostasis in the body is violated, which leads to reproductive and metabolic disorders, which leads higher-producing cows to have a lower number of lactations. 

Disinfection both before and after milking was determined to be a statistically significant risk factor for BTM samples testing positive for *M. bovis*. Farms employing disinfection before milking had 19.5 times greater odds of testing positive for *M. bovis* in BTM samples (*p* < 0.009), whereas those disinfecting after milking had 14.59 times greater odds of testing positive for *M. bovis* in BTM samples (*p* < 0.002). This is an interesting finding and may be associated with the breed of cows, farm size, high lactation, and milk management. Similar findings were observed in a Swiss study where the milking process was evaluated as a risk factor for the presence of *M. bovis* in milk samples [39]. 

According to the average milk yield per farm for positive BTM samples, we concluded that cows with milk yield above 8000 L are at the greatest risk of excreting *M. bovis* in milk, in accordance with the previously published results of [2]. These results indicate that higher-producing cows are at a higher risk for *M. bovis* infection. The frequency of *M. bovis* detection in the BTM samples was attributed to several risk factors, and these findings highlight the significance of biosecurity precaution and enhanced hygiene for decreasing the risk of *M. bovis* contamination of bulk tanks.

## 5. Conclusions

Based on its findings, this study has revealed, for the first time, the common occurrence of *M. bovis* in dairy farms in Serbia. The identified risk factors associated with the presence of *M. bovis* in dairy cow farms encompass a broad spectrum of farm management practices, herd types, facility maintenance protocols, and milking management, emphasising the multifactorial nature of these conditions. In conclusion, we suggest that the bigger the production and the farm, the greater the risk, which often contradicts the biosecurity measures that can be improved in larger systems. Our PCR results underscore the significance of monitoring BTM milk analyses on dairy farms to identify and mitigate the spread of *M. bovis*. The potential transmission of *M. bovis* through milk presents a concern for the dissemination and progression of respiratory illnesses, particularly among calves.

## Figures and Tables

**Table 1 pathogens-13-00302-t001:** Overview of results of collected samples and farms.

Outcome	PCR BTM (*n* = 115)	PCR NS (*n* = 61)	ELISA Farm (*n* = 61)	ELISA Cow (*n* = 307)
+	11 (9.57)	3 (4.92%)	14 (22.95%)	116 (37.79%)
−	104 (90.43%)	58 (95.08%)	47 (77.05%)	191 (62.21%)

**Table 2 pathogens-13-00302-t002:** Distribution of positive *M. bovis* DNA of BTM dairy farms according to herd size.

Number of Cows	Total Farms	Positive Farms of BTM	(%)	Negative Farms of BTM	(%)
0–20	79	0	0%	79	100%
21–200	23	4	17.39%	19	82.61%
201–	13	7	71.43%	2	28.57%
Total	115	11	9.57%	104	90.43%

**Table 3 pathogens-13-00302-t003:** Distribution of positive *M. bovis* DNA of BTM farms according to milk yield.

Milk Yield	Total Farms	Positive Farms of BTM	(%)	Negative Farms of BTM	(%)
<6000	67	0	0%	67	100%
6001–8000	30	4	13.33%	26	87.67%
8001+	18	7	38.89%	11	61.11%
Total	115	11	9.57%	104	90.43%

**Table 4 pathogens-13-00302-t004:** Summary of the continuous variables showing the number of herds (n) and the median and interquartile range (IQR) for herds.

	*M. bovis*-Positive BTM	Valid	Median	Mean	Std. Deviation	Minimum	Maximum	IRQ	*p*
Number of cows at farm	No	104	9	54.952	146.19	1	1000	18	<0.001
	Yes	11	350	349.455	273.69	34	850	427	
Milk yield	No	104	5800	6193.75	1396.66	4500	10,800	1300	<0.001
	Yes	11	8600	8581.82	1361.48	6100	10,200	1750	
Number of cows for bulk milk	No	104	9	54.952	146.19	1	1000	18	<0.001
	Yes	11	150	216.091	177.71	34	600	219	
Parity	No	56	5	5.518	1.84	3	11	2	0.044
	Yes	5	4	3.8	0.44	3	4	0	

**Table 5 pathogens-13-00302-t005:** Summary of risk factors in association with bulk tank milk PCR-positive results for *M. bovis*.

		BTM				95% Confidence Intervals		
	Categories	No	Yes	Total	Odds Ratio	Lower	Upper	*p*
Breed	HF	20	9	29	0.053	0.011	0.264	<0.001
	Sim	84	2	86				
Positive nasal swab	No	53	5	58	1.982	0.084	46.734	1
	Yes	3	0	3				
Bacterial mastitis	No	24	0	24	8.292	0.437	157.197	0.147
	Yes	32	5	37				
Seropositive farms	No	47	0	47	55	2.8	1080.325	<0.001
	Yes	9	5	14				
Disinfection before milking	No	52	2	54	19.5	2.49	152.694	0.009
	Yes	4	3	7				
Disinfection after milking	No	32	0	32	14.592	0.77	276.616	0.02
	Yes	24	5	29				
Type of farm	Corporate	2	5	7	0.004	0	0.098	<0.001
	Family	54	0	54				
Type of milking	Machine	36	5	41	0.162	0.009	3.078	0.162
	Manual	20	0	20				
Type of holding	Tie stall	53	4	57	4.417	0.37	52.787	0.296
	Free stall	3	1	4				
Overcrowding	No	51	3	54	6.8	0.91	50.81	0.096
	Yes	5	2	7				
Inadequate amb. con.	No	46	3	49	3.067	0.452	20.822	0.252
	Yes	10	2	12				

**Table 6 pathogens-13-00302-t006:** Regression analysis table of risk factors for *M. bovis*-positive BTM samples.

				95% Confidence Interval	
	Coefficients	Odds Ratio	*p*	Lower Bound	Upper Bound
(Intercept)	−2.605	0.074	0.068		
Overcrowding (Yes)	−1.696	0.183	0.257	0.01	3.450
Breed (HF)	4.175	65.069	0.002	4.925	859.662

**Table 7 pathogens-13-00302-t007:** Values of the relative risks in different milk yields of cows tested for the presence of *M. bovis* DNA in BTM samples.

Milk Yield	Relative Risk	Confidence Limits		*p*-Value
		Lower Limit	Upper Limit	
8001+ vs. <8000	9.43	3.07	28.93	0.0001
8001+ vs. >6000–8000	2.92	0.99	8.59	0.0738

**Table 8 pathogens-13-00302-t008:** Values of the relative risk for *M. bovis* infection depending on the herd size.

Herd Size	Relative Risk	Confidence Limits		*p*-Value
		Lower Limit	Upper Limit	
51+ vs. <50	16.2	3.74	70.23	0.0002
101+ vs. <100	11.94	3.45	41.24	0.0001
201+ vs. >200	13.70	4.64	40.61	0.0001
301+ vs. >300	10.30	3.69	28.71	0.0001

## Data Availability

Data available on request from the authors.

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
