# Peer review of "Presence of Mycoplasma bovis in Bulk Tank Milk and Associated Risk Factor Analysis in Serbian Dairy Farms"

_pathogens, 2024, doi:10.3390/pathogens13040302_

Round 1
Reviewer 1 Report
Comments and Suggestions for Authors
The publication is focused on presence or “prevalence” of Mycoplasma bovis in bulk tank milk samples using the conventional ELISA test and PCR, risk factors were also analysed in Serbian dairy farms. However, there are some major issues that require attention before the manuscript can be accepted.
Major:
I believe that the statistical analysis is not appropriate to present the results obtained, the use of other statistical methods that have more power to explain the dependent variable through the risk factors, one of them could be the multivariate logistic regression analysis and use the odds ratio to explain if there are differences between risk factors.
L84: The authors could provide the questionnaire used in the study as a supplementary file.
Minor:
L169: replace “LL” by “low limited”
L181: replace “Table 7” by “(Table 7)
- The PCR test values (positivity) were associated with the risk factors, and because they were not associated with the higher prevalence obtained with the ELISA test.
Comments on the Quality of English Language
Minor editing of English language required
Author Response
Reviewer 1:
We wish to thank you all for your constructive comments in this review. Your comments provided valuable insights to refine its contents and analysis. In this document, we try to address the issues raised as best as possible.
The publication is focused on presence or “prevalence” of Mycoplasma bovis in bulk tank milk samples using the conventional ELISA test and PCR, risk factors were also analysed in Serbian dairy farms. However, there are some major issues that require attention before the manuscript can be accepted.
Major:
I believe that the statistical analysis is not appropriate to present the results obtained, the use of other statistical methods that have more power to explain the dependent variable through the risk factors, one of them could be the multivariate logistic regression analysis and use the odds ratio to explain if there are differences between risk factors.
Author’s Reply: We supplemented and expanded the statistical analysis of risk factors as per your suggestion. Risk factors of M. bovis infection were studied through a mixed effects logistic regression model. Please see, the manuscript.
L84: The authors could provide the questionnaire used in the study as a supplementary file.
Author’s Reply: We provide the questionnaire as a supplementary file.
Minor:
L169: replace “LL” by “low limited”
Author’s Reply: We replaced and changed in Tables.
L181: replace “Table 7” by “(Table 7)
Author’s Reply: We accepted the recommendation. Now (Table 8).
- The PCR test values (positivity) were associated with the risk factors, and because they were not associated with the higher prevalence obtained with the ELISA test.
Comments on the Quality of English Language: Minor editing of English language required
Author’s Reply: The English was reviewed by native English speaker, we can provide certificate.
Reviewer 2 Report
Comments and Suggestions for Authors
The present study presents the prevalence of M. bovis in dairy farms located in the region of Serbia. In my opinion it’s a well written manuscript, that gives many details on how the study was conducted and analyses sufficiently the results. There are no data about M. bovis in Serbia or the broader geographic area, thus the present study adds information in the literature. Below I mention some observations.
L. 50: results is preferable to be presented in introduction
L. 71-72: Ethical approval must be at the end of Material and methods, or before references
L. 197: missing the study.
L. 292: missing the study.
Author Response
Reviewer 2
We wish to thank you all for your constructive comments in this review. Your comments provided valuable insights to refine its contents and analysis. In this document, we try to address the issues raised as best as possible
Comments and Suggestions for Authors
The present study presents the prevalence of M. bovis in dairy farms located in the region of Serbia. In my opinion it’s a well written manuscript, that gives many details on how the study was conducted and analyses sufficiently the results. There are no data about M. bovis in Serbia or the broader geographic area, thus the present study adds information in the literature. Below I mention some observations.
- 50: results is preferable to be presented in introduction
Author’s Reply: We changed the sentence. The results of the study by Gille et al. presented in the discussion and discussed accordingly.
- 71-72: Ethical approval must be at the end of Material and methods, or before references
Author’s Reply: We relocated ethical approval before references.
- 197: missing the study.
Author’s Reply: We added the reference according per your suggestion.
- 292: missing the study
Author’s Reply: L 292 is data availability statement.
Note: We supplemented and expanded the statistical analysis of risk factors at the request of reviewer 1.
Round 2
Reviewer 1 Report
Comments and Suggestions for Authors
The authors have responded to each of the reviewers´ comments and suggestions.